# Cold dispase digestion of murine lungs improves recovery and culture of airway epithelial cells

Piotr Pawel Janas[1], Caroline Chauché[1], Patrick Shearer[2], Georgia Perona-Wright[2], Henry J. McSorley[3], Jürgen Schwarze[1] *

1 Centre for Inflammation Research, Institute for Regeneration and Repair, The University of Edinburgh, Edinburgh BioQuarter, Edinburgh, United Kingdom, 2 Institute of Infection, Immunity & Inflammation, University of Glasgow, Glasgow, United Kingdom, 3 Division of Cell Signalling and Immunology, School of Life Sciences, University of Dundee, Dundee, United Kingdom

* Jurgen.Schwarze@ed.ac.uk

**Data Availability Statement:** Data is now available from Edinburgh Data Share public data repository with assigned DOI - https://doi.org/10.7488/ds/7506.

## Abstract

Airway epithelial cells (AECs) play a key role in maintaining lung homeostasis, epithelium regeneration and the initiation of pulmonary immune responses. To isolate and study murine AECs investigators have classically used short and hot (1h 37˚C) digestion protocols. Here, we present a workflow for efficient AECs isolation and culture, utilizing long and cold (20h 4˚C) dispase II digestion of murine lungs. This protocol yields a greater number of viable AECs compared to an established 1h 37˚C dispase II digestion. Using a combination of flow cytometry and immunofluorescent microscopy, we demonstrate that compared to the established method, the cold digestion allows for recovery of a 3-fold higher number of CD45⁻CD31⁻EpCAM⁺ cells from murine lungs. Their viability is increased compared to established protocols, they can be isolated in larger numbers by magnetic-activated cell sorting (MACS), and they result in greater numbers of distal airway stem cell (DASC) KRT5⁺p63⁺ colonies *in vitro*. Our findings demonstrate that temperature and duration of murine lung enzymatic digestion have a considerable impact on AEC yield, viability, and ability to form colonies *in vitro*. We believe this workflow will be helpful for studying lung AECs and their role in the biology of lung.

## Introduction

The airway epithelium forms an important barrier at the interface of environment and organism [1]. AECs consist of heterogenous groups of cells, including pneumocytes, goblet, club, ciliated and basal cells [2, 3]. Depending on the species and anatomic location in the airways, different proportions of AEC subsets have been observed. The epithelium of large airways in humans forms a pseudostratified layer with ciliated, goblet and club cells at the luminal side, while underlying basal cells are attached to the basement membrane. On the other hand, the pseudostratified epithelium in mice is limited to the trachea, with simpler columnar epithelium present in the bronchi [1]. It is thought that in mice the majority of airway basal cells are

**Funding:** PJ - PHD16-19 BUSH - British Lung Fundation - https://statistics.blf.org.uk/ - NO CC - VET/2020 -1 EPDF 7 - Horserace Betting Levy Board - https://www.hblb.org.uk/ - NO.

**Competing interests:** The authors have declared that no competing interests exist.

limited to the trachea, with sparse basal cells in bronchi [1]. Among AECs, basal cells keep the greatest degree of pluripotency allowing them to proliferate and differentiate into various sub-types of epithelial cells. Thus, basal cells play a major role in airway epithelium homeostasis and repair [2, 3] and they are also essential for *in vitro* primary AECs cultures since they can be maintained in a constant proliferative state [4–6]. However, another rare type of pluripotent airway epithelial progenitor cell, a distal airway stem cell (DASC) characterised by co-expression of p63 and Krt5 was described to play a substantial role in alveolar regeneration following injury in both humans and mice [3, 7, 8].

Isolated primary AECs are used in a range of *in vitro* experimental systems. Most commonly, when maintained in media containing differentiation inhibitors, primary AECs can be subcultured in a monolayer. However, when cultured in well-plate inserts with specialised media, air-liquid interface (ALI) cultures can be established [4]. ALI cultures recapitulate the epithelium of large airways as these cultures are pseudostratified and AECs differentiate into various subsets including ciliated, club or goblet cells. ALI cultures also exhibit mucociliary motion, produce mucus and develop tight junctions [9–11]. Subsets of AECs can also be cultured in a Matrigel matrix to generate tracheospheres, bronchiolospheres or alveolospheres [12, 13]. All these experimental systems can be used to study AEC biology at homeostasis, after injury e.g. by respiratory pathogens, and during regeneration.

In recent years, it has been recognised that AECs do not only form an inert barrier, but that they are a dynamic and versatile cell population that is involved in maintaining lung homeostasis [14], mounting initial immune responses [15] and lung regeneration following injury [16]. The manifold roles of AECs implicate them in pathogenesis of a number of diseases ranging from asthma [17], COPD [18] to lung fibrosis [19]. An increasing number of research questions focusing on AECs in various contexts creates a demand for improved and well-defined methodologies allowing for their efficient harvest, isolation and culture.

In order to study primary AECs *in vitro*, various digestion techniques have been employed that allow for disruption of cell junctions and connective tissue resulting in lung single cell suspension. To isolate and study AECs from a murine lung, researchers commonly utilize a combination of dispase II and DNase I enzymes at 37˚C for 30-60min [20–22] (hot digestion). However, when we employed this established digestion method, we found that the yield and viability of AECs was unsatisfactory in the context of MACS sorting and *in vitro* culture. We therefore investigated alternative workflows. While there are several digestion methods for murine lungs utilizing a range of enzymes including collagenases, dispases, liberase, pronase and trypsin [23], we investigated the effects of digestion time and temperature rather than the type of enzyme. It has been shown that long, overnight digestion can be beneficial to yield and efficiency when isolating cells from human skin [24], or human brain tissue [25]. Following on from a method where murine tracheas were digested for 18h at 4˚C using pronase, however without benchmarking of cell yield against other frequently employed methods [9], we assessed a 20h 4˚C dispase II/DNase I digestion of murine lungs (cold digestion). We coupled presented hot and cold digestions with a two-step MACS sorting that allows for depletion of CD45$^+$ and CD31$^+$ and enrichment of EpCAM$^+$ cells. The initial depletion step ensures that none of the EpCAM-expressing macrophages [26] are enriched together with epithelial cells.

Here, we present a workflow that in comparison to the commonly used hot dispase digestion allows for retrieval of a 3-fold higher number of CD45$^-$CD31$^-$EpCAM$^+$ AECs with a higher proportion of viable cells. This in turn allows for recovery of greater numbers of AECs after MACS sorting and a greater yield of keratin 5 (KRT5)$^+$ p63$^+$ DASC colonies *in vitro* after seven-day culture.

## Materials and methods

### Animals

C57BL/6 mice were bred and housed at the University of Edinburgh. Mice were housed in individually ventilated cages. All procedures approved by the University of Edinburgh Animal Welfare and Ethical Review Board, and performed under UK Home Office licenses with institutional oversight performed by qualified veterinarians. UK Home Office project license to JS, number P4871232F. ARRIVE guidelines were followed.

### Murine lung harvest and digestion

Adult C57BL/6 mice over six weeks of age were anaesthetized with a 1:1 mixture of ketamine and medetomidine through intraperitoneal (IP) injection with a dose dependent on the body weight. Cervical dislocation was carried out, and death confirmed by terminal exsanguination. The abdominal aorta was cut, and the trachea exposed by removing the salivary glands. The trachea was then cannulated using a blunt needle and bronchoalveolar lavage with pre-warmed Dulbecco's Modified Eagle Medium/Nutrient Mixture F-12 (DMEM/F12) (Gibco) was performed (three times 0.8ml per mouse). The rib cage was then removed, and cardiovascular system was flushed by injecting 10ml of ice-cold DMEM/F12 into the right ventricle. A volume of 1.5-2ml of enzyme mix was then slowly injected through the tracheal cannula into the lungs, which were then re-moved (lungs were dissected out of the thoracic cavity by cutting at the primary bronchial bifurcation) and placed in 3ml of digestion mixture (2mg/ml Dispase II (Sigma-Aldrich) + 0.1mg/ml DNaseI (Sigma-Aldrich) re-suspended in DMEM/F12 (Gibco) + 1% v/v Penicillin-Streptomycin (Gibco)). The digestion mixture was previously filtered through 0.22μm filter and frozen. Lungs were then incubated for 1h at 37˚C or 20h at 4˚C. The digestion solution together with lungs was then poured onto a 70μm strainer, lungs were then dissociated using a 5ml syringe plunger by gently agitating the lungs on the mesh. The strainer was then washed using 10ml of dispase wash (DMEM/F12 (Gibco) + 0.05mg/ml DNaseI (Sigma-Aldrich) + 1% v/v Penicillin-Streptomycin (Gibco)). Each sample was then centrifuged for 15min, 4˚C, 130 relative centrifugal force (RCF). The supernatant was then removed and 2ml of ice-cold ACK red blood cell lysis (Gibco) was added to each tube, samples were swirled for 90s, and 5ml of MACS buffer (Phosphate-buffered saline (PBS)—no $Mg^{2+}$ and $Ca^{2+}$ + 0.5% bovine serum albumin (BSA) + 2mM ethylenediaminetetraacetic acid (EDTA)) + 1% v/v Penicillin/Streptomycin (10,000 U/ml, Gibco) was added. Samples were then passed through a 40μm strainer, washed with 5ml of MACS buffer and centrifuged for 5min, 4˚C at 300 RCF. Each sample was then resuspended in 1ml of MACS buffer and blocked with 5μl of 0.5mg/ml anti-mouse CD16/32 antibody (BioLegend) ready for further MACS processing or flow cytometry staining.

### Airway epithelial cell sorting (MACS)

After incubation cells were centrifuged for 5min, 4˚C, 300 RCF and resuspended in 85μl of MACS buffer + 5μl of anti-CD31 microbeads and 10μl of anti-CD45 microbeads (Miltenyi Biotec) per $10^7$ total cells. Samples were then incubated for 30 minutes on ice. Each sample was then washed with 1ml/$10^7$ cells of MACS buffer and centrifuged for 5min, 4˚C at 300 RCF. Cells were then passed through LS MACS columns (Miltenyi Biotec) on a QuadroMACS separator (Miltenyi Biotec) and flowthrough was collected. Collected cells were then centrifuged for 5min, 4˚C, 300 RCF and resuspended in left-over liquid, and 15μl of anti-EpCAM microbeads (Miltenyi Biotec) was added. Samples were incubated on ice for 30 minutes, washed with 1ml MACS buffer, and centrifuged for 5min, 4˚C, 300 RCF. Supernatant was

discarded, and cells were resuspended in 500μl of MACS buffer. The cell suspension was then passed through a MS MACS column using an OctoMACS separator (Miltenyi Biotec). EpCAM⁺ cells were then gently flushed out of the column by applying 2ml of MACS buffer and inserting the plunger into the column. Cells were then centrifuged for 5min at 4°C, 300 RCF, supernatant was removed, and the cell pellet was resuspended in 0.5ml of supplemented airway epithelial growth media. 10μl of cell suspension was then aspirated and gently mixed with 10μl of 0.1% trypan blue (Gibco) by pipetting up and down. Immediately afterwards, 10μl of cell suspension was placed in Neubauer improved counting chamber (haemocytometer) and covered with coverslip. Cells were then manually counted and total amount of live (trypan blue negative) and dead (trypan blue positive) cells in suspension was calculated.

## Primary AECs *in vitro* culture

24-well plates were coated with a coating solution that combines 30μg/ml of type I calf skin collagen (Sigma-Aldrich), 10μg/ml human placental fibronectin (Bio-Techne) and 10μg/ml bovine serum albumin (BSA–Sigma-Aldrich) in HBS (Gibco), for at least 4-8h at 37°C. A total of 2x10$^5$ MACS sorted AECs were seeded per well and 0.5ml media was changed after 24h, and then every two days thereafter. Media were prepared by combining supplemented Promo-Cell Epithelial Growth Medium with 1μM A 83–01 (STEMCELL Technologies) and 0.2μM DMH1 (STEMCELL Technologies) SMAD signalling inhibitors, 5μM Y-27632 ROCK (STEMCELL Technologies) pathway inhibitor, 0.5μM CHIR99021 (STEMCELL Technologies) WNT pathway activator and 1% v/v Penicillin/Streptomycin (Gibco). Cells were cultured for seven days, followed by immunofluorescent microscopy of each well.

The protocol described in this peer-reviewed article is published on protocols.io (https://dx.doi.org/10.17504/protocols.io.rm7vzxo68gx1/v1) and is included for printing purposes as S1 File.

## Immunofluorescent microscopy

Cells in each well were washed with PBS three times and fixed using 4% PFA solution for 10min at room temperature (RT). Fixed cells were then stored in 70% Ethanol at 4°C until staining. Cells were permeabilised with a 1% BSA and 0.2% triton X-100 (Sigma-Aldrich) solution at RT for 10min. Cells were then blocked for 45min at RT using a 5% goat serum, 1% BSA and 0.1% Tween-20 (Sigma-Aldrich) in PBS. Cells were then stained with a mix of primary antibodies (anti-KRT5, anti-p63 and anti-E-Cadherin-eF660), overnight at 4°C, followed by three washes with blocking buffer (each wash for 5min) and staining with secondary antibodies (anti-rabbit-AF488, anti-mouse-AF555) as well as 1:500 10mg/ml Hoechst (Invitrogen) for nuclear counter stain for 1h at RT (Table 1). Secondary antibodies were then washed away

**Table 1. Immunofluorescent microscopy antibodies.**

| Antibody name | Supplier | Host species | Antibody type | Clone | Catalogue number | Antibody Registry ID | Dilution |
|---|---|---|---|---|---|---|---|
| Anti-p63 | Abcam | Mouse | Monoclonal | 4A4 | ab735 | AB_305870 | 1:200 |
| Anti-KRT5 | BioLegend | Rabbit | Polyclonal | Poly19055 | 905503 | AB_2734679 | 1:500 |
| Anti-E-cadherin (CD324) eFluor 660 | Thermo Fisher Scientific | Rat | Monoclonal | DECMA-1 | 50-3249-82 | AB_11040003 | 1:30 |
| Rat IgG1 kappa Isotype Control (eBRG1) eFluor 660 | Thermo Fisher Scientific | Rat | Isotype control | eBRG1 | 50-4301-82 | AB_10598505 | 1:30 |
| Goat Anti-Rabbit IgG (H+L) Alexa Fluor 488 | Thermo Fisher Scientific | Goat | Polyclonal, secondary | Reactivity—rabbit | A-11008 | AB_143165 | 1:200 |
| Goat Anti-Mouse IgG (H&L) Alexa Fluor 555 | Abcam | Goat | Polyclonal—secondary | Reactivity—mouse | ab150114 | AB_2687594 | 1:200 |

with three washes (each wash for 5min) with blocking buffer followed by filling each well with 0.7ml PBS.

Imaging was performed using EVOS FL Auto 2 (Thermo Fisher Scientific), using 20x objective as well as DAPI, GFP and RFP light cubes. Scans were set to a custom calibrated Corning 24-well plate, where 25% of each well was scanned from the middle using the "more overlap" setting and manually set focus for each channel. Images were analysed using ImageJ, with ICA LUT (look-up table) set for KRT5 for easier basal AECs identification. All micrograph related colony counting was performed while blinded.

Surface area of KRT5 colonies was calculated using ImageJ. At first KRT5 images were converted to 8-bit. Then images were smoothed out using median filter (7px value), followed by thresholding using "mean setting". Images were then converted to binary format and surface area of black pixels was calculated using "Analyze particles" tool. Obtained pixel numbers were then converted to total surface in mm2 based on image metadata (scale).

## Flow Cytometry

Following digestion, numbers of cells in each sample were calculated using a haemocytometer. $0.5 \times 10^6$ total cells from each sample were then washed twice with PBS and stained using a LIVE/DEAD fixable near-IR stain (Invitrogen) in PBS for 30min on ice, followed by two washes with PBS and staining with a combination of anti-CD45, anti-CD31 and anti-EpCAM in PBS + 1% BSA for 30min on ice (Table 2). Stained samples were then washed twice with PBS + 1% BSA. Whole samples were acquired and unmixed using Cytek Aurora with Cytek SpectroFlo 3.0.3 and analysed using De Novo Software FCSexpress 7.

The gating strategy involved debris exclusion (side scatter/forward scatter–SSC-H/FSC-H), followed by selection for singlets (FSC-H/FSC-A) and dead cell exclusion (LIVE/DEAD Fixable Near IR/FSC-H). Dead cell exclusion was performed at the end of gating strategy for viability evaluation in Fig 2B. Leukocytes and endothelial cells were then excluded using CD45/CD31 gate, followed by selection for AECs with EpCAM/SSC-H gate.

## RNA isolation and qPCR (quantitative polymerase chain reaction)

After $CD45^-CD31^-EpCAM^+$ MACS sorting cells were centrifuged for 5min, 4°C, 300 RCF and resuspended in 0.5ml of TRizol. Following 5min incubation at RT, samples were stored at -80°C before further processing. Samples were then thawed and 100µl of bromochloropropane (Sigma-Aldrich) was added per 500µl of TRizol. Samples were shaken vigorously for 30s until they acquired a milky-pink shade. Samples were then incubated at RT for 10min, and then cooled down on ice for 5min before centrifugation at 4°C, 16,000 RCF for 20min. 200µl of the top aqueous phase was then transferred to fresh Eppendorf test tubes and 250µl of RNase-free isopropanol (Merck) and 1µl of GlycoBlue Coprecipitant (Invitrogen) were added. Test tubes

Table 2. Flow cytometry antibodies.

| Antibody name | Supplier | Host species | Antibody type | Clone | Catalogue number | Antibody Registry ID | Dilution |
|---|---|---|---|---|---|---|---|
| Anti-CD45 Pacific Blue | BioLegend | Rat | Monoclonal | S18009F | 157212 | AB_2876534 | 1:200 |
| Anti-CD45 AF700 | BioLegend | Rat | Monoclonal | S18009F | 157210 | AB_2860730 | 1:200 |
| Anti-CD31 BV605 | BioLegend | Rat | Monoclonal | 390 | 102427 | AB_2563982 | 1:600 |
| Anti-CD31 BV421 | BioLegend | Rat | Monoclonal | 390 | 102423 | AB_2562186 | 1:300 |
| Anti-EpCAM PE/Dazzle594 | BioLegend | Rat | Monoclonal | G8.8 | 118236 | AB_2632777 | 1:300 |
| Anti-EpCAM BV605 | BioLegend | Rat | Monoclonal | G8.8 | 118227 | AB_2563984 | 1:300 |
| Anti-CD24 PE/Cyanine7 | BioLegend | Rat | Monoclonal | M1/69 | 101821 | AB_756048 | 1:200 |
| Anti-CD49f BV605 | BioLegend | Rat | Monoclonal | GoH3 | 313625 | AB_2616782 | 1:100 |

were inverted ten times and incubated at RT for 10min, followed by centrifugation at 4°C, 16,000 RCF for 10min. Blue RNA pellet could now be observed at the bottom of test tubes. Supernatant was removed and pellet was washed three times using 1ml of RNase-free 75% EtOH (Merck), with 5min 4°C, 16,000 RCF centrifugations between each wash. After final wash, ethanol was discarded, and pellets were air-dried until no liquid could be seen in test tubes. Pellets were then resuspended in 30μl of RNAse-free water (QIAGEN), and amount of RNA was quantified using NanoDrop One/OneC Microvolume UV-Vis Spectrophotometer (Thermo Scientific). The amount of RNA was normalised in each sample using RNase-free water and 300ng of RNA was converted to cDNA using High-Capacity cDNA Reverse Transcription Kit (Applied Biosystems). qPCRs were performed in duplicates using Fast SYBR Green Master Mix (Applied Biosystems) and StepOne (Thermo Fisher Scientific) thermocycler. 100nM of forward and reverse primers, as well as 10ng of cDNA, were added to each qPCR reaction (Table 3). Rpl37 was used as an endogenous control. Genomic DNA contamination was assessed by performing a qPCR reaction on non-reverse transcribed RNA sample.

## Statistical analysis

Data in the plots are represented as medians with boxes representing interquartile ranges and bars corresponding to minima and maxima. MFI is reported as median fluorescence intensity. D'Agostino & Pearson test was used to assess the normality of data distribution. Statistical differences were assessed using a parametric unpaired two-tailed t-test. All outliers were included. Data in the text is presented as means with standard error. All statistical analysis was performed using GraphPad Prism 8.4.3.

## Results

### Cold digestion provides greater yield and viability of AECs compared to hot digestion

We used flow cytometry (Fig 1A) to assess the yield of three different cell populations after a 1h 37°C dispase II/DNase I digestion of murine lungs. On average, we obtained $5.55 \pm 0.05$ x$10^5$ CD45$^-$CD31$^-$EpCAM$^+$ AECs (7% of total cells), $7.35 \pm 0.59$ x$10^6$ CD45$^+$ immune cells and $2.10 \pm 0.03$ x$10^5$ CD31$^+$ endothelial cells (Fig 1B).

We decided to investigate whether the use of cold digestion of murine lungs would increase the yield and/or viability of AECs. Using the cold digestion approach, the average number of isolated CD45$^-$CD31$^-$EpCAM$^+$ AECs increased to $1.46 \pm 0.15$ x$10^6$, almost 3-fold the number obtained following hot digestion. Likewise, the number of CD31$^+$ endothelial cells obtained with the cold digestion ($4.27 \pm 0.05$ x$10^5$) was almost 2-fold higher than after hot digestion. Conversely, the number of CD45$^+$ cells was similar between the two digestion techniques. This suggests that the cold digestion approach is beneficial for recovery of structural cells from the lung, while obtaining similar numbers of lung immune cells.

While several cell surface markers like CD4, CD8 or PD-1L are known to be sensitive to dispase II digestion [27], we did not find any changes in expression levels of CD45, CD31 and EpCAM between the digestion methods as seen by comparable MFI (median) levels of each marker irrespective of digestion method (Fig 1C). We also did not observe loss of several AECs-related cell surface markers such as major histocompatibility (MHC) I, MHC-II or CD24 (S1 Fig).

### Following cold digestion, more AECs are viable compared to hot digestion

Next, we compared the viability of isolated AECs between cold and hot lung digestions by flow cytometry, assessing the proportion of live cells within the CD45$^-$CD31$^-$EpCAM$^+$ population

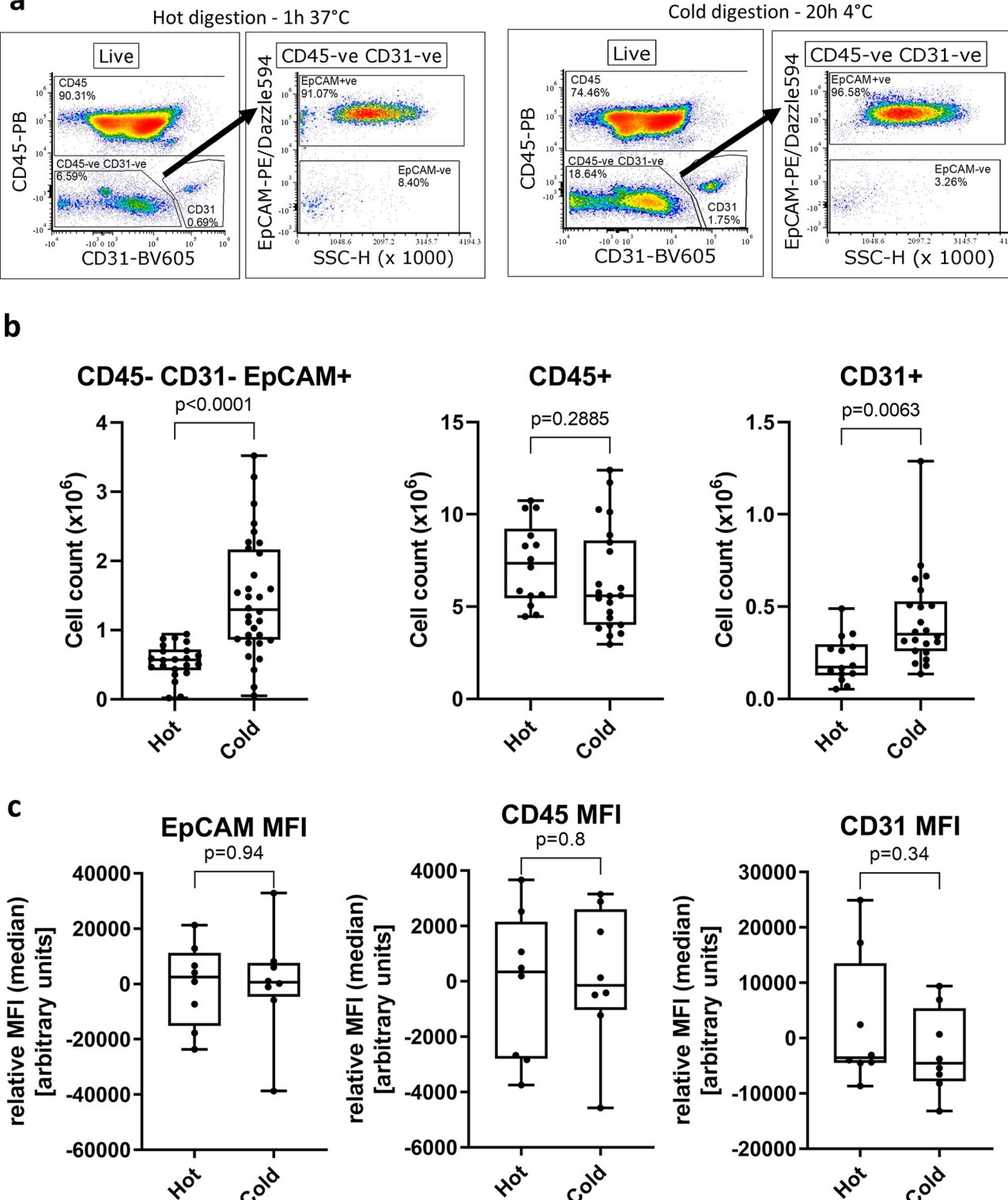

**Fig 1. Flow cytometric analysis of murine lung digests with lung inflation, comparing the number of cells after hot (1h, 37°C) or cold (20h, 4°C) digests.** (**a**) Representative gating strategy following debris exclusion (FSC/SSC), singlets (FSC-H/FSC-A) and dead cells exclusion (LIVE/DEAD fixable near-IR). CD45⁻CD31⁻ cells are gated, followed by EpCAM⁺ gating. (**b**) Number of CD45⁻CD31⁻EpCAM⁺ AECs, CD31⁺ endothelial cells or CD45⁺ leukocytes. (**c**) Comparison of relative MFI (median) values for EpCAM, CD45 and CD31 between cold and hot digests within respective CD45⁻CD31⁻EpCAM⁺, CD45⁺ and CD31⁺ gates. Each data point represents a lung from a single mouse. Unpaired t-test, median ± min/max.

in single cell suspension, with the gating strategy shown in Fig 2A. We found that on average 83.99 ± 1.16% of CD45$^-$CD31$^-$EpCAM$^+$ cells from cold lung digestions were alive, compared to 56.65 ± 3.16% after hot digestion, a 1.48-fold increase in viable AECs (Fig 2B). Despite differences in viability of AECs, we did not observe changes in levels of Ldha (a marker of oxidative stress) [28] between digestion methods (S2 Fig).

Given this increased viability, we hypothesised that MACS sorting, and subsequent seeding of AECs isolated by cold digestion would result in a greater number of basal AEC colonies in *in vitro* cultures, compared to hot digestion. Performing double MACS sorting with initial CD45 and CD31 depletion followed by EpCAM positive selection, we recovered substantially more AECs following cold digestion (5.30 ± 0.45 x10$^5$ AECs) compared to hot digestion (2.97 ± 0.26 x10$^5$ AECs) which is equivalent to a 1.78-fold increase in number of viable AECs (Fig 2C). The AECs recovered after MACS sorting were highly pure (Fig 2D), with average purity of 96.35 ± 0.75% CD45$^-$CD31$^-$EpCAM$^+$ cells of live cells. To confirm the airway epithelial identity of sorted CD45$^-$CD31$^-$EpCAM$^+$ we used flow cytometry to confirm that these cells also express other epithelial markers. Virtually all CD45$^-$CD31$^-$EpCAM$^+$ cells also expressed CD49f (integrin α6) [29, 30] and CD24 [31, 32] (Fig 2E), albeit as expected, at vastly varying levels.

### Following cold digestion and MACS sorting, more basal cell colonies proliferate *in vitro*

To investigate if the observed increase in AEC viability with cold digestion translated to greater recovery of airway basal cells *in vitro*, we seeded 2x10$^5$ MACS sorted CD45$^-$CD31$^-$EpCAM$^+$ cells (98% purity) from hot or cold lung digestions into 24-well plates (workflow indicated in Fig 3A). Cells were cultured for seven days in Promocell airway epithelial media supplemented with differentiation inhibitors and a Wnt pathway activator [4, 33, 34]. After seven days the cells were fixed and stained for markers of epithelial (E-cadherin) and basal cells (KRT5 and p63) [1, 35]. Then 25% of the surface area of each 24-well plate was imaged and the number of basal cell colonies was quantified, as well as the total surface area of KRT5$^+$ cell colonies per well (Fig 3B). KRT5$^+$ colony morphology was visualized using phase-contrast microscopy (Fig 3C). On average there were 21 ± 2.97 basal cell colonies after hot digestion and 34 ± 2.98 colonies after cold digestion per 52.6mm$^2$, which equals to a 1.62-fold increase in colony counts. When we assessed the total surface area of KRT5$^+$ colonies between the two digestion methods, we found a 2.3-fold increase in colony surface area with an average surface area occupied by KRT5$^+$ colonies of 0.66 ± 0.13mm$^2$ after cold digestions compared to 0.30 ± 0.07mm$^2$ after hot digestions (Fig 3D). Furthermore, there were very few E-cadherin negative non-epithelial cells in cultures irrespective of the digestion method employed, suggesting that the combination of cold digestion, MACS sorting, and appropriate media allows for propagating highly pure AECs.

## Discussion

Here, we demonstrate that by changing standard dispase II/DNase I digestion conditions from 1h 37°C to 20h 4°C we are able to obtain significantly greater numbers of lung structural cells, including AECs. This, together with the observation that a higher proportion of AECs are viable after cold digestion suggests that there is a potential to employ cold dispase II lung digestion for establishing primary murine AECs cultures *in vitro*. Indeed, in combination with MACS CD45$^-$CD31$^-$EpCAM$^+$ AEC sorting, we demonstrate that culture of AECs after cold dispase digestion results in a significantly greater number of DASC colonies with a larger surface area than after hot digestion. Although establishing culture protocols for murine lung

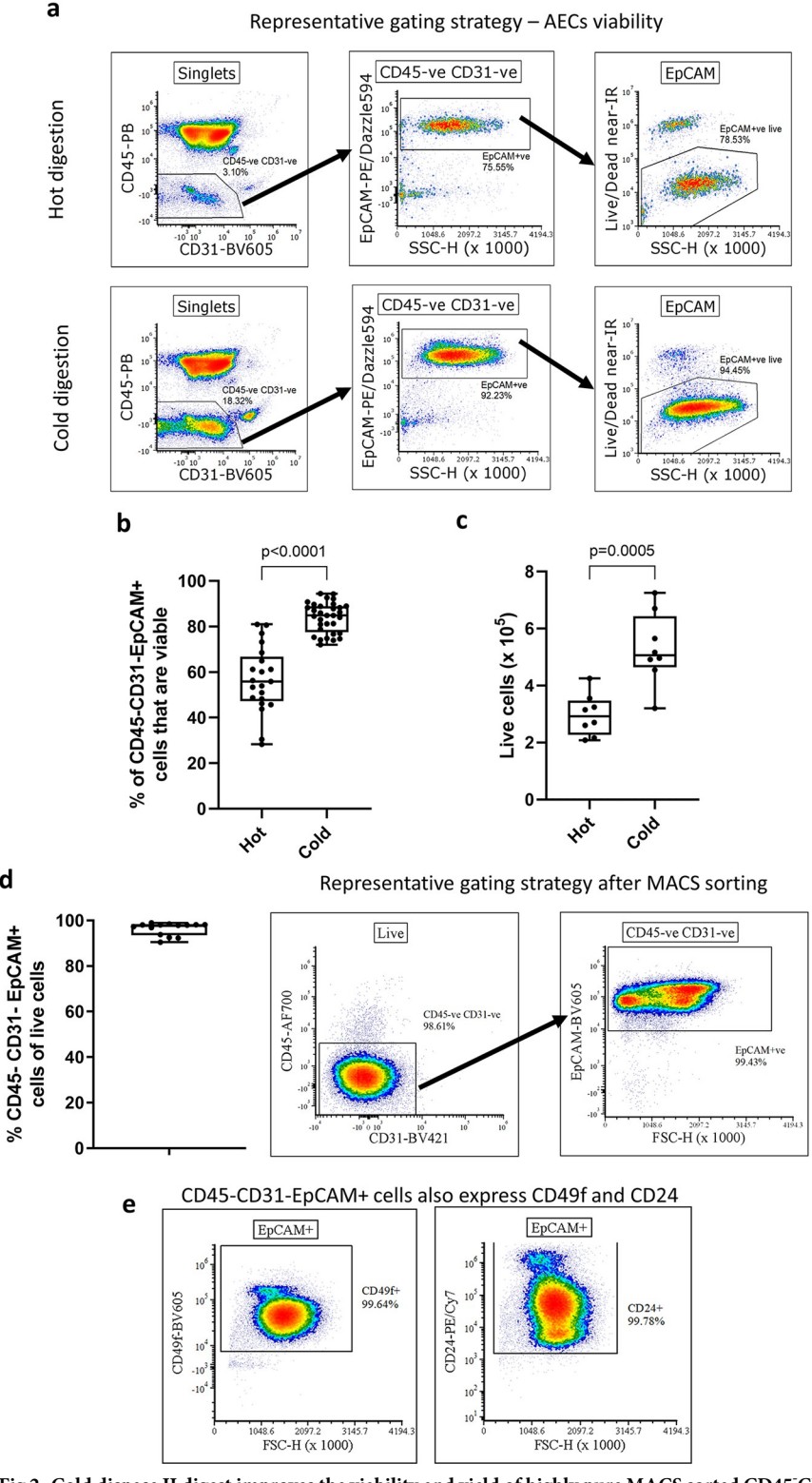

**Fig 2. Cold dispase II digest improves the viability and yield of highly pure MACS sorted CD45⁻CD31⁻EpCAM⁺ cells.** (**a**) Representative gating strategy for hot and cold digestion following debris exclusion (FSC/SSC) and singlets (FSC-H/FSC-A) for analysis of viability of CD45⁻CD31⁻EpCAM⁺ AECs. (**b**) Comparison of frequency of viable FACS-sorted CD45⁻CD31⁻EpCAM⁺ through live/dead near-IR fixable dye. Unpaired t-test (n = 21–33), median ± min/max. (**c**) Comparison of cell yield between the digest methods after CD45, CD31 MACS depletion and EpCAM⁺ MACS

selection. Quantification using haemocytometer and trypan blue live/dead exclusion. Unpaired t-test (n = 8), median ± min/max. (**d**) Evaluation of cell suspension purity after MACS sorting CD45⁻CD31⁻EpCAM⁺ cells using flow cytometry. Each data point corresponds to a single murine lung. (**e**) Validating the CD45⁻CD31⁻EpCAM⁺ cells identity. CD45⁻CD31⁻EpCAM⁺ cells also express CD49f and CD24. Presented gating followed debris exclusion (FSC/SSC), singlets gating (FSC-H/FSC-A), live cells gating (LIVE/DEAD fixable near-IR/FSC-H), CD45 and CD31 exclusion (CD45/CD31) and EpCAM⁺ gating (EpCAM/FSC-H).

airway epithelial progenitors was not within the scope of this project, by altering media composition we were able to culture CD45⁻CD31⁻EpCAM⁺ MACS sorted cells for up to 14 days over two passages (S3A Fig). Additionally, all cells did express KRT5 at the end of passaging period (S3B Fig).

These improvements overcome the need for pooled AEC isolates from several mice [9, 32, 36] and allow for a more comprehensive analysis of AECs from individual animals. This will contribute to the principles of replacement, reduction, and refinement (3R) [37] by reducing the number of mice used in AEC research. However, the greater yield and viability are not the only benefits. The fact that after cold digestion the number of isolated AECs increases substantially suggests that overall AEC populations, including rare and difficult-to-isolate populations, may be better represented in comparison to hot digestion. While further analysis is required to formally confirm our prediction, this is likely to have major implications for researchers studying transcriptomics, epigenomics or proteomics of isolated murine AECs, especially at single cell level. It is possible that by employing hot digestion, only the mostly easily dislodged and/or the most robust AECs are released and survive, potentially skewing analysis. When comparing the abundance of cell types in the murine lung between single-cell RNA sequencing (scRNA-seq) and single-nuclear RNA sequencing (snRNAseq), it was reported that scRNAseq under-represents epithelial population in comparison to snRNAseq. Indeed, neuroendocrine and basal cells were exclusively seen in the snRNAseq data sets, while type-1 pneumocytes, club cells and ciliated cells were much better represented in scRNAseq. This might be explained by the fact that single-cell suspension for the scRNAseq is obtained through enzymatic digestion, while nuclei for snRNAseq are isolated through physical dissociation of the tissue [38]. This observation therefore suggests that, in order to isolate representative live epithelial populations through enzymatic digestion, further optimisation is required.

A method commonly used in combination with AEC isolation from tracheas is the removal of fibroblasts by adherence, which requires an additional incubation step, which may further contribute to deteriorating the viability of AECs [9, 39]. Alternatively, FACS is used to sort AECs from lung digestions, however, FACS may not only be detrimental to viability of sorted cells, but also limits the number of samples that can be sorted in parallel [40, 41]. Here we suggest combining the cold digestion approach with a two-step MACS sorting of CD45⁻CD31⁻EpCAM⁺ cells, which allows for parallel sorting of multiple highly pure samples from larger animal cohorts.

Historically, researchers used tracheas to isolate basal cells from murine airways [4, 9], however, the workflow presented here allows for isolation of sufficient numbers of viable airway

**Table 3. List of qPCR primers.**

| Primer | Sequence |
|---|---|
| *Ldha* forward | CATTGTCAAGTACAGTCCACACT |
| *Ldha* reverse | TTCCAATTACTCGGTTTTTGGGA |
| *Rpl37* forward | CCAAGCGCAAGAGGAAGTATAAC |
| *Rpl37* reverse | GAATCCATGTCTGAATCTGCGG |

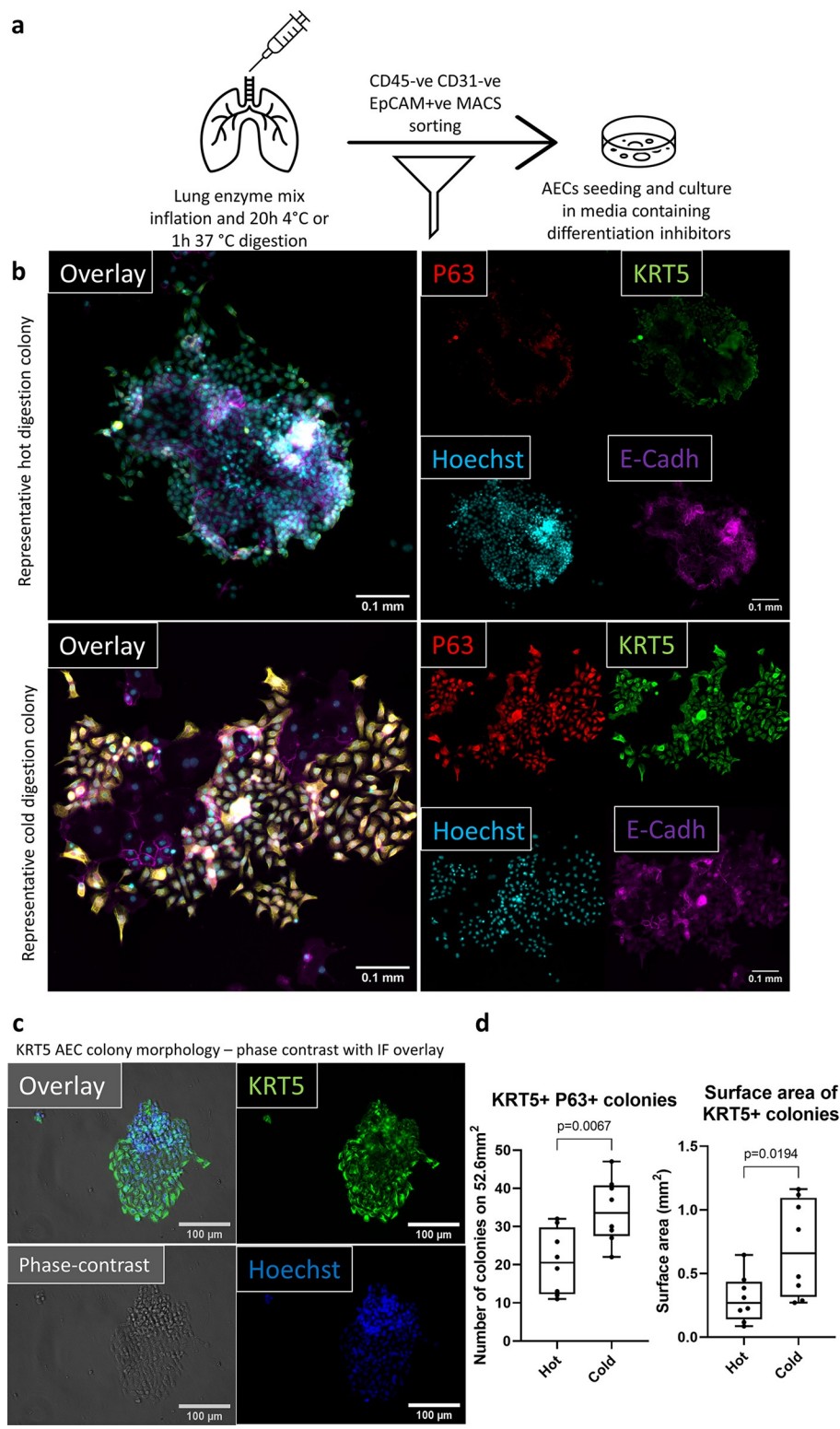

**Fig 3. Improved AECs viability translates to a better performance in *in vitro* culture. Cold digestion culture yields a greater number of basal cell colonies, which occupy a larger area.** (**a**) Workflow diagram representing the steps involved in digesting the murine lungs and MACS sorting AECs for *in vitro* culture. (**b**) Representative immunofluorescent micrographs (cold and hot digestions) of P63 (red), KRT5 (green), Hoechst (blue) and E-Cadherin (magenta) used for quantification of the number of P63 and KRT5 double positive colonies. Scale bar 0.1mm. 25% of

each 24-WP well imaged, which equals to area of 52.6mm². Each well was imaged using 20x objective starting from the centre. (**c**) 20x magnification phase-contrast (grey) image overlayed with immunofluorescent KRT5 (green) and Hoechst (blue) images to present morphology of KRT5⁺ cells following cold digestion. Scale bar (white bar), 100μm (**d**) Quantification of the number (left) and surface area (right) of basal AECs colonies. Colony was defined as ten E-cadh/P63/KRT5⁺ cells in immediate vicinity. Colony numbers were quantified based on E-cadh/P63/KRT5⁺ cells, while colony surface was quantified using only KRT5 pixels. Unpaired t-test (n = 8), median ± min/max. Each data point is a single well cultured from MACS sorted CD45⁻CD31⁻ EpCAM⁺ isolated from a single murine lung.

epithelial progenitor cells from the lung (without trachea) for successful *in vitro* cultures. These lung-derived progenitor cells differ from their tracheal counterparts [42], and culture models based on lung derived basal cells may allow a closer representation of the pulmonary epithelium.

In conclusion, we have established a workflow which is based on cold dispase II/DNase I digestion that allows for recovery of a greater number of highly viable AECs, which together with MACS sorting enables quick and parallel isolation of highly pure AEC populations from murine lungs that are suitable for *in vitro* culture.

## Supporting information

**S1 Fig. Cold digestion with dispase II does not remove MHC-I, MHC-II or CD24 from the surface of CD45⁻CD31⁻EpCAM⁺ cells.** Following debris exclusion (FSC/SSC), singlets gating (FSC-H/FSC-A), live cells gating (LIVE/DEAD fixable near-IR/FSC-H), CD45 and CD31 exclusion (CD45/CD31) and EpCAM⁺ gating (EpCAM/FSC-H) expression of MHC-I, MHC-II and CD24 was evaluated. Gating was based on fluorescence minus one (FMO) controls.
(TIF)

**S2 Fig. Digestion type does not affect cellular oxidative stress based on LDHa (lactate dehydrogenase) levels.** Following either hot or cold digestion CD45⁻CD31⁻EpCAM⁺ cells were MACS sorted and RNA isolated using Trizol and chloroform-based RNA isolation. SYBR green qPCR was performed, with Rpl37a as endogenous control. N = 3–4.
(TIF)

**S3 Fig. Murine AECs can be passaged (P0-2) following cold digestion and CD45⁻CD31⁻EpCAM⁺ MACS sorting.** (a) Following cold digestion, murine AECs (CD34⁻CD31⁻EpCAM⁺) were MACS sorted and 2x10⁵ cells were seeded into a 24-well plate and cultured in a Stemcell PneumaCult Ex-Plus medium with 10μM Y-27632, 3μM CHIR99021, 1μM A 83–01 for 14 days. Cells were passaged twice. Each time cells were detached using TrypLE and split 1:10 into a larger well plate starting from 24-well plate (P0), followed by 12-well plate (P1) and lastly a 6-well plate (P2). Once cells reached confluency at the end of passage 2, cells were lifted up using TrypLE and 1.4x10⁶ live cells (trypan blue exclusion) were counted using haemocytometer. Six 10x objective phase-contrast single fields of view (3x2) were stitched together. Scale bar 400μm. N>2. (b) All cells express KRT5 (green) after three passages of CD45⁻CD31⁻EpCAM⁺ MACS sorted AECs. Scale bar, 300μm.
(TIF)

**S1 Protocol. Step-by-step protocol including materials and recipes.**
(DOCX)

**S1 File. Step-by-step protocol, also available on protocols.io.** https://dx.doi.org/10.17504/protocols.io.rm7vzxo68gx1/v1.
(PDF)

## Acknowledgments

PPJ and JS thank the Breathing Together Consortium for their support. We gratefully acknowledge assistance and expertise from the Bioresearch and Veterinary staff, Flow Core Facility and Microscopy Core Facility at the University of Edinburgh.

## Author Contributions

**Conceptualization:** Piotr Pawel Janas, Caroline Chauché, Henry J. McSorley, Jürgen Schwarze.

**Data curation:** Piotr Pawel Janas, Jürgen Schwarze.

**Formal analysis:** Piotr Pawel Janas, Caroline Chauché, Patrick Shearer, Georgia Perona-Wright, Henry J. McSorley.

**Funding acquisition:** Jürgen Schwarze.

**Investigation:** Piotr Pawel Janas, Caroline Chauché, Henry J. McSorley.

**Methodology:** Piotr Pawel Janas.

**Project administration:** Jürgen Schwarze.

**Supervision:** Jürgen Schwarze.

**Visualization:** Piotr Pawel Janas.

**Writing – original draft:** Piotr Pawel Janas, Jürgen Schwarze.

**Writing – review & editing:** Piotr Pawel Janas, Caroline Chauché, Patrick Shearer, Georgia Perona-Wright, Henry J. McSorley, Jürgen Schwarze.

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
