## [Decision Letter · Decision Letter 0]

18 Sep 2023

PONE-D-23-26951Cold dispase digestion of murine lungs improves recovery and culture of airway epithelial cellsPLOS ONE

Dear Dr. Schwarze,

Thank you for submitting your manuscript to PLOS ONE. After careful consideration, we feel that it has merit but does not fully meet PLOS ONE’s publication criteria as it currently stands. Therefore, we invite you to submit a revised version of the manuscript that addresses the points summarized below raised during the review process.

We look forward to receiving your revised manuscript.

Kind regards,

Dominique Heymann, Ph.D.

Academic Editor

PLOS ONE

3. We note you have not yet provided a protocols.io PDF version of your protocol and/or a protocols.io DOI. When you submit your revision, please provide a PDF version of your protocol as generated by protocols.io (the file will have the protocols.io logo in the upper right corner of the first page) as a Supporting Information file. The filename should be S1_file.pdf, and you should enter “S1 File” into the Description field. Any additional protocols should be numbered S2, S3, and so on. Please also follow the instructions for Supporting Information captions [https://journals.plos.org/plosone/s/supporting-information#loc-captions]. The title in the caption should read: “Step-by-step protocol, also available on protocols.io.”

Please assign your protocol a protocols.io DOI, if you have not already done so, and include the following line in the Materials and Methods section of your manuscript: “The protocol described in this peer-reviewed article is published on protocols.io (https://dx.doi.org/10.17504/protocols.io.[...]) and is included for printing purposes as S1 File.” You should also supply the DOI in the Protocols.io DOI field of the submission form when you submit your revision.

If you have not yet uploaded your protocol to protocols.io, you are invited to use the platform’s protocol entry service [https://www.protocols.io/we-enter-protocols] for doing so, at no charge. Through this service, the team at protocols.io will enter your protocol for you and format it in a way that takes advantage of the platform’s features. When submitting your protocol to the protocol entry service please include the customer code PLOS2022 in the Note field and indicate that your protocol is associated with a PLOS ONE Lab Protocol Submission. You should also include the title and manuscript number of your PLOS ONE submission.

4. Please include captions for your S4 at the end of your manuscript, and update any in-text citations to match accordingly. Please see our Supporting Information guidelines for more information: http://journals.plos.org/plosone/s/supporting-information. 

5. . We note that Figure 1a, 2a, 2d, 2e, 3a, 3b, 3c, S1 and S3 in your submission contain copyrighted images. All PLOS content is published under the Creative Commons Attribution License (CC BY 4.0), which means that the manuscript, images, and Supporting Information files will be freely available online, and any third party is permitted to access, download, copy, distribute, and use these materials in any way, even commercially, with proper attribution. For more information, see our copyright guidelines: http://journals.plos.org/plosone/s/licenses-and-copyright.

A. You may seek permission from the original copyright holder of Figure 1a, 2a, 2d, 2e, 3a, 3b, 3c, S1 and S3 to publish the content specifically under the CC BY 4.0 license. 

B. If you are unable to obtain permission from the original copyright holder to publish these figures under the CC BY 4.0 license or if the copyright holder’s requirements are incompatible with the CC BY 4.0 license, please either i) remove the figure or ii) supply a replacement figure that complies with the CC BY 4.0 license. Please check copyright information on all replacement figures and update the figure caption with source information. If applicable, please specify in the figure caption text when a figure is similar but not identical to the original image and is therefore for illustrative purposes only.

Reviewers' comments:

Reviewer's Responses to Questions

**Comments to the Author**

1. Does the manuscript report a protocol which is of utility to the research community and adds value to the published literature?

Reviewer #1: Yes

2. Has the protocol been described in sufficient detail?

To answer this question, please click the link to protocols.io in the Materials and Methods section of the manuscript (if a link has been provided) or consult the step-by-step protocol in the Supporting Information files.

The step-by-step protocol should contain sufficient detail for another researcher to be able to reproduce all experiments and analyses.

Reviewer #1: Partly

3. Does the protocol describe a validated method?

Reviewer #1: Yes

4. If the manuscript contains new data, have the authors made this data fully available?

Reviewer #1: N/A

**5. Is the article presented in an intelligible fashion and written in standard English?**

Reviewer #1: Yes

6. Review Comments to the Author

Reviewer #1: In this work, authors described and compared protocols to isolate and study murine airway epithelial cells (AECs). In particular, they investigated the effects of digestion time and temperature and they showed that a long and cold (20h 4°C) dispase II digestion of murine lungs have a considerable positive impact on AEC yield, viability, and ability to form colonies in vitro compared to an established 1h 37°C dispase II digestion step. Finally, they proposed a modified workflow for efficient AECs isolation that improves recovery and culture of airway epithelial cells. To contribute to the principles of replacement, reduction, and refinement (3R) in AEC research, isolated primary AECs are used in a range of in vitro experimental systems. The workflow presented in this manuscript may therefore interest a wide range of readers studying lung AECs and their role in the biology of lung. The manuscript is clear and well-written. Nevertheless, the authors should perform the minor modifications below to be suitable for publication.

1- The introduction could describe why you have chosen a magnetic-based isolation of cells with the depletion of CD45pos and CD31pos cells before the positive selection of EpCAM-expressing cells.

2- Please homogenize the writing of CD45-CD31-EpCAM+. Sometimes you used CD45-CD31-EpCAM+ and CD45-CD31EpCAM+.

In Materials and Methods

3- In the “Murine lung harvest and digestion” part, line 94 : what does mean (S4)?

4- In the “Primary AECs in vitro culture” part, please verify the description of the medium between lines 144 and 147. See also supplement S4 protocol part. For example, line 144, “Media was prepared” should be replaced by ““Media were prepared” ; line 145, 1µM A3801 does not seem correct. Do you mean #03801 medium ? What is 1µM ? ; line 146, CHIR9902 should be CHIR 99021.

5- In “immunofluorescent microscopy” part, line 155 and line 157, “an-ti” should be replaced by “anti”.

6- In “RNA isolation and qPCR” part, line 193 and line 194, “ul” should be replaced by “µl”.

In the legends of Figure

7- In Fig 3b, you mentioned “Representative immunofluorescent micrographs (cold digestion)” line 308, but you have also shown results for hot digestion, please modify.

8- What is the magnification used for the images ?

9- What is “Top scale bar, 2mm”, line 310?

10- In Fig 3c, could you specify if the images concern cold or hot digestion condition ?

In supplement S4 protocol

11- “Militenyi” should be replaced by “Miltenyi”.

12- In “recipes”, “4C°” should be replaced by “4°C”.

13- Please check step 28. Is the cell pellet resuspended in 100µl of airway epithelial growth media (as indicated in supplement S4 protocol) or 0.5ml of MACS buffer (as indicated in Materials and Methods).

14- Please add the coating step description before step 30.

7. PLOS authors have the option to publish the peer review history of their article (what does this mean?). If published, this will include your full peer review and any attached files.

Reviewer #1: No

---

## [Author Response · Author response to Decision Letter 0]

31 Oct 2023

We would like to thank you and the reviewer for your constructive comments. We have now fully revised the paper as detailed point-by-point below.

Editor

We revised the manuscript and confirm that it is prepared according to PLOS ONE’s style requirements as found in the pdf files shared by the editor. We also revised the file naming requirements and confirm that they follow the PLOS ONE’s guidelines. 

The data present in the manuscript is now uploaded and available at Edinburgh DataShare public repository. The data can be accessed via https://doi.org/10.7488/ds/7506

3. We note you have not yet provided a protocols.io PDF version of your protocol and/or a protocols.io DOI. When you submit your revision, please provide a PDF version of your protocol as generated by protocols.io (the file will have the protocols.io logo in the upper right corner of the first page) as a Supporting Information file. The filename should be S1_file.pdf, and you should enter “S1 File” into the Description field. Any additional protocols should be numbered S2, S3, and so on. Please also follow the instructions for Supporting Information captions [https://journals.plos.org/plosone/s/supporting-information#loc-captions]. The title in the caption should read: “Step-by-step protocol, also available on protocols.io.” Please assign your protocol a protocols.io DOI, if you have not already done so, and include the following line in the Materials and Methods section of your manuscript: “The protocol described in this peer-reviewed article is published on protocols.io (https://dx.doi.org/10.17504/protocols.io.[...]) and is included for printing purposes as S1 File.” You should also supply the DOI in the Protocols.io DOI field of the submission form when you submit your revision. If you have not yet uploaded your protocol to protocols.io, you are invited to use the platform’s protocol entry service [https://www.protocols.io/we-enter-protocols] for doing so, at no charge. Through this service, the team at protocols.io will enter your protocol for you and format it in a way that takes advantage of the platform’s features. When submitting your protocol to the protocol entry service please include the customer code PLOS2022 in the Note field and indicate that your protocol is associated with a PLOS ONE Lab Protocol Submission. You should also include the title and manuscript number of your PLOS ONE submission.

The protocol is now uploaded the protocol to protocols.io. It can be accessed via dx.doi.org/10.17504/protocols.io.rm7vzxo68gx1/v1 . We are also enclosing the protocols.io pdf version of the protocol as S1_file.pdf. The caption for the S1_file can now be found at the end of manuscript.

4. Please include captions for your S4 at the end of your manuscript, and update any in-text citations to match accordingly. Please see our Supporting Information guidelines for more information: http://journals.plos.org/plosone/s/supporting-information. 

The captions are now included for the S4 at the end of the manuscript, with updated in-text citations.

5. We note that Figure 1a, 2a, 2d, 2e, 3a, 3b, 3c, S1 and S3 in your submission contain copyrighted images. All PLOS content is published under the Creative Commons Attribution License (CC BY 4.0), which means that the manuscript, images, and Supporting Information files will be freely available online, and any third party is permitted to access, download, copy, distribute, and use these materials in any way, even commercially, with proper attribution. For more information, see our copyright guidelines: http://journals.plos.org/plosone/s/licenses-and-copyright. We require you to either (1) present written permission from the copyright holder to publish these figures specifically under the CC BY 4.0 license, or (2) remove the figures from your submission: A. You may seek permission from the original copyright holder of Figure 1a, 2a, 2d, 2e, 3a, 3b, 3c, S1 and S3 to publish the content specifically under the CC BY 4.0 license. We recommend that you contact the original copyright holder with the Content Permission Form (http://journals.plos.org/plosone/s/file?id=7c09/content-permission-form.pdf) and the following text: “I request permission for the open-access journal PLOS ONE to publish XXX under the Creative Commons Attribution License (CCAL) CC BY 4.0 (http://creativecommons.org/licenses/by/4.0/). Please be aware that this license allows unrestricted use and distribution, even commercially, by third parties. Please reply and provide explicit written permission to publish XXX under a CC BY license and complete the attached form.” Please upload the completed Content Permission Form or other proof of granted permissions as an ""Other"" file with your submission. In the figure caption of the copyrighted figure, please include the following text: “Reprinted from [ref] under a CC BY license, with permission from [name of publisher], original copyright [original copyright year].”B. If you are unable to obtain permission from the original copyright holder to publish these figures under the CC BY 4.0 license or if the copyright holder’s requirements are incompatible with the CC BY 4.0 license, please either i) remove the figure or ii) supply a replacement figure that complies with the CC BY 4.0 license. Please check copyright information on all replacement figures and update the figure caption with source information. If applicable, please specify in the figure caption text when a figure is similar but not identical to the original image and is therefore for illustrative purposes only.

All figures are now part of submission to data repository Edinburgh Data Share (https://doi.org/10.7488/ds/7506) , and as such all the figures are already under the CC BY 4.0 license. Please see https://datashare.ed.ac.uk/bitstream/handle/10283/8524/license_text?sequence=2&isAllowed=y for license details. 

The reference list was reviewed and we confirm that none of the included citations were retracted to the best of our knowledge. 

Reviewer #1

1. The introduction could describe why you have chosen a magnetic-based isolation of cells with the depletion of CD45pos and CD31pos cells before the positive selection of EpCAM-expressing cells.

We now explain in the introduction (lines 80-83) why we are performing a two-step MACS sorting of EpCAM cells. 

2. Please homogenize the writing of CD45-CD31-EpCAM+. Sometimes you used CD45-CD31-EpCAM+ and CD45-CD31EpCAM+.

The writing of CD45-CD31-EpCAM+ was corrected and unified across the manuscript. 

3. In the “Murine lung harvest and digestion” part, line 94 : what does mean (S4)?

The S4 refers to supplemental information 4 – the protocol that is now submitted to protocols.io per PLOS ONE guidelines. As editor requested for a sentence in materials and methods referring to protocols.io protocol, the brackets are now removed. 

4. In the “Primary AECs in vitro culture” part, please verify the description of the medium between lines 144 and 147. See also supplement S4 protocol part. For example, line 144, “Media was prepared” should be replaced by ““Media were prepared” ; line 145, 1µM A3801 does not seem correct. Do you mean #03801 medium ? What is 1µM ? ; line 146, CHIR9902 should be CHIR 99021.

We apologies for the typos. These were now corrected (updated lines 147-150). 

5. In “immunofluorescent microscopy” part, line 155 and line 157, “an-ti” should be replaced by “anti”.

Thank you for pointing these out. These are now corrected. Corrected updated lines 162 and 164.

6. In “RNA isolation and qPCR” part, line 193 and line 194, “ul” should be replaced by “µl”.

7. Thank you for pointing these out. These are now corrected. Corrected updated lines 197 and 198.

8. In Fig 3b, you mentioned “Representative immunofluorescent micrographs (cold digestion)” line 308, but you have also shown results for hot digestion, please modify.

Thank you for pointing that out. The figure description now indicates both hot and cold digestion. Corrected updated lines 311.

9. What is the magnification used for the images ?

The 20x magnification is now indicated in updated lines 317-318. 

10. What is “Top scale bar, 2mm”, line 310?

Thank you for pointing that out. The “Top scale bar, 2mm” is now removed. The figure description was corrected and indicates a single scale bar of 0.1mm. Updated lines 316. 

11. In Fig 3c, could you specify if the images concern cold or hot digestion condition?

The Fig 3c description now specifies that it concerns cold digestion condition. Updated line 320. 

12. “Militenyi” should be replaced by “Miltenyi”.

Thank you for pointing that out. The typo was corrected throughout the manuscript and S4 supplemental protocol. 

13. In “recipes”, “4C°” should be replaced by “4°C”.

The typo was corrected. Thank you for pointing it out.

14. Please check step 28. Is the cell pellet resuspended in 100µl of airway epithelial growth media (as indicated in supplement S4 protocol) or 0.5ml of MACS buffer (as indicated in Materials and Methods).

Thank you for pointing out the discrepancy. Both the S4 protocol and the manuscript were corrected to indicate 0.5ml of supplemented airway epithelial growth media. Updated line 136.

15. Please add the coating step description before step 30.

Following step is now present in the protocol: “30. Coat the well plate with well-plate coating solution and incubate for at least 4-8h at 37°C before use.”

We warmly thank the Reviewer for their positive appraisal, which together with their constructive comments and suggestions have greatly helped us in submitting an improved manuscript to your Journal. We hope this will be received positively and look forward to your decision in due course.

---

## [Decision Letter · Decision Letter 1]

14 Dec 2023

PONE-D-23-26951R1Cold dispase digestion of murine lungs improves recovery and culture of airway epithelial cellsPLOS ONE

Dear Dr. Schwarze,

Thank you for submitting your manuscript to PLOS ONE. After careful consideration, we feel that it has merit but does not fully meet PLOS ONE’s publication criteria as it currently stands. Therefore, we invite you to submit a revised version of the manuscript that addresses the points raised during the review process.

We look forward to receiving your revised manuscript.

Kind regards,

Dominique Heymann, Ph.D.

Academic Editor

PLOS ONE

Journal Requirements:

Reviewers' comments:

Reviewer's Responses to Questions

**Comments to the Author**

1. Does the manuscript report a protocol which is of utility to the research community and adds value to the published literature?

Reviewer #1: Yes

2. Has the protocol been described in sufficient detail?

To answer this question, please click the link to protocols.io in the Materials and Methods section of the manuscript (if a link has been provided) or consult the step-by-step protocol in the Supporting Information files.

The step-by-step protocol should contain sufficient detail for another researcher to be able to reproduce all experiments and analyses.

Reviewer #1: Yes

3. Does the protocol describe a validated method?

Reviewer #1: Yes

4. If the manuscript contains new data, have the authors made this data fully available?

Reviewer #1: N/A

**5. Is the article presented in an intelligible fashion and written in standard English?**

Reviewer #1: Yes

6. Review Comments to the Author

Reviewer #1: There are still some discrepancies in your protocols. Please modify.

1- Chemical product names are still not correct. Please check thoroughly all your files (manuscript, supplementary…).

CHIR99021 from stemcell Tech should not be CHIR9902 or CHIR9901 (reviewed manuscript).

A 83-01 from stemcell Tech should not be A8301 in S3 Fig (manuscript, line 391).

2- Do you really have 1% Pen/Strep in your MACS buffer (S4 protocol) ? It is not indicated in “Murine lung harvest and digestion” part (manuscript, line 118).

In addition, in “Recipes” of S4 protocol, you should name this buffer “MACS buffer” instead of “MACS buffer WASH”. It is confusing.

3- In “Recipes” of S4 protocol, you did correct as requested the “4C⁰” in “Airway epithelial growth media“ part, but you have forgotten to correct the “4C⁰” in “Well-plate coating solution “ part. Please modify.

4- Thanks for adding as requested the coating step (step 30) in S4 protocol. However, it would have been better if you had not made a typo. “coate” should be replaced by “coat”.

7. PLOS authors have the option to publish the peer review history of their article (what does this mean?). If published, this will include your full peer review and any attached files.

Reviewer #1: No

---

## [Author Response · Author response to Decision Letter 1]

5 Jan 2024

To 

Dr Dominique Heymann

 04.01.2024

Dear Dr Heymann,

Manuscript submission

PONE-D-23-26951

“Cold dispase digestion of murine lungs improves recovery and culture of 

airway epithelial cells” 

We would like to thank you and the reviewer for your constructive comments. We have now fully revised the paper as detailed point-by-point below. Our comments are in blue. 

Editor

We revised the manuscript and confirm that it is prepared according to PLOS ONE’s style requirements as found in the pdf files shared by the editor. We also revised the file naming requirements and confirm that they follow the PLOS ONE’s guidelines. 

The data present in the manuscript is now uploaded and available at Edinburgh DataShare public repository. The data can be accessed via https://doi.org/10.7488/ds/7506

3. We note you have not yet provided a protocols.io PDF version of your protocol and/or a protocols.io DOI. When you submit your revision, please provide a PDF version of your protocol as generated by protocols.io (the file will have the protocols.io logo in the upper right corner of the first page) as a Supporting Information file. The filename should be S1_file.pdf, and you should enter “S1 File” into the Description field. Any additional protocols should be numbered S2, S3, and so on. Please also follow the instructions for Supporting Information captions [https://journals.plos.org/plosone/s/supporting-information#loc-captions]. The title in the caption should read: “Step-by-step protocol, also available on protocols.io.” Please assign your protocol a protocols.io DOI, if you have not already done so, and include the following line in the Materials and Methods section of your manuscript: “The protocol described in this peer-reviewed article is published on protocols.io (https://dx.doi.org/10.17504/protocols.io.[...]) and is included for printing purposes as S1 File.” You should also supply the DOI in the Protocols.io DOI field of the submission form when you submit your revision. If you have not yet uploaded your protocol to protocols.io, you are invited to use the platform’s protocol entry service [https://www.protocols.io/we-enter-protocols] for doing so, at no charge. Through this service, the team at protocols.io will enter your protocol for you and format it in a way that takes advantage of the platform’s features. When submitting your protocol to the protocol entry service please include the customer code PLOS2022 in the Note field and indicate that your protocol is associated with a PLOS ONE Lab Protocol Submission. You should also include the title and manuscript number of your PLOS ONE submission.

The protocol is now uploaded the protocol to protocols.io. It can be accessed via dx.doi.org/10.17504/protocols.io.rm7vzxo68gx1/v1 . We are also enclosing the protocols.io pdf version of the protocol as S1_file.pdf. The caption for the S1_file can now be found at the end of manuscript.

4. Please include captions for your S4 at the end of your manuscript, and update any in-text citations to match accordingly. Please see our Supporting Information guidelines for more information: http://journals.plos.org/plosone/s/supporting-information. 

The captions are now included for the S4 at the end of the manuscript, with updated in-text citations.

5. We note that Figure 1a, 2a, 2d, 2e, 3a, 3b, 3c, S1 and S3 in your submission contain copyrighted images. All PLOS content is published under the Creative Commons Attribution License (CC BY 4.0), which means that the manuscript, images, and Supporting Information files will be freely available online, and any third party is permitted to access, download, copy, distribute, and use these materials in any way, even commercially, with proper attribution. For more information, see our copyright guidelines: http://journals.plos.org/plosone/s/licenses-and-copyright. We require you to either (1) present written permission from the copyright holder to publish these figures specifically under the CC BY 4.0 license, or (2) remove the figures from your submission: A. You may seek permission from the original copyright holder of Figure 1a, 2a, 2d, 2e, 3a, 3b, 3c, S1 and S3 to publish the content specifically under the CC BY 4.0 license. We recommend that you contact the original copyright holder with the Content Permission Form (http://journals.plos.org/plosone/s/file?id=7c09/content-permission-form.pdf) and the following text: “I request permission for the open-access journal PLOS ONE to publish XXX under the Creative Commons Attribution License (CCAL) CC BY 4.0 (http://creativecommons.org/licenses/by/4.0/). Please be aware that this license allows unrestricted use and distribution, even commercially, by third parties. Please reply and provide explicit written permission to publish XXX under a CC BY license and complete the attached form.” Please upload the completed Content Permission Form or other proof of granted permissions as an ""Other"" file with your submission. In the figure caption of the copyrighted figure, please include the following text: “Reprinted from [ref] under a CC BY license, with permission from [name of publisher], original copyright [original copyright year].”B. If you are unable to obtain permission from the original copyright holder to publish these figures under the CC BY 4.0 license or if the copyright holder’s requirements are incompatible with the CC BY 4.0 license, please either i) remove the figure or ii) supply a replacement figure that complies with the CC BY 4.0 license. Please check copyright information on all replacement figures and update the figure caption with source information. If applicable, please specify in the figure caption text when a figure is similar but not identical to the original image and is therefore for illustrative purposes only.

All figures are now part of submission to data repository Edinburgh Data Share (https://doi.org/10.7488/ds/7506) , and as such all the figures are already under the CC BY 4.0 license. Please see https://datashare.ed.ac.uk/bitstream/handle/10283/8524/license_text?sequence=2&isAllowed=y for license details. 

The reference list was reviewed, and we confirm that as of 14/12/2023 none of the included citations were retracted based on information available on PubMed. Several references (6, 11, 32) were corrected so that they indicate an exact article, rather than a collection of articles. One reference (27) was added as a consequence of edits suggested by reviewer #1. 

Thank you for your suggestion to use PACE. We confirm that all figures were uploaded to PACE to make sure that they meet PLOS requirement. 

Reviewer #1

1. The introduction could describe why you have chosen a magnetic-based isolation of cells with the depletion of CD45pos and CD31pos cells before the positive selection of EpCAM-expressing cells.

We now explain in the introduction (lines 80-83) why we are performing a two-step MACS sorting of EpCAM cells. 

2. Please homogenize the writing of CD45-CD31-EpCAM+. Sometimes you used CD45-CD31-EpCAM+ and CD45-CD31EpCAM+.

The writing of CD45-CD31-EpCAM+ was corrected and unified across the manuscript. 

3. In the “Murine lung harvest and digestion” part, line 94 : what does mean (S4)?

The S4 refers to supplemental information 4 – the protocol that is now submitted to protocols.io per PLOS ONE guidelines. As editor requested for a sentence in materials and methods referring to protocols.io protocol, the brackets are now removed. 

4. In the “Primary AECs in vitro culture” part, please verify the description of the medium between lines 144 and 147. See also supplement S4 protocol part. For example, line 144, “Media was prepared” should be replaced by ““Media were prepared” ; line 145, 1µM A3801 does not seem correct. Do you mean #03801 medium ? What is 1µM ? ; line 146, CHIR9902 should be CHIR 99021.

We apologies for the typos. These were now corrected (updated lines 147-150). 

5. In “immunofluorescent microscopy” part, line 155 and line 157, “an-ti” should be replaced by “anti”.

Thank you for pointing these out. These are now corrected. Corrected updated lines 162 and 164.

6. In “RNA isolation and qPCR” part, line 193 and line 194, “ul” should be replaced by “µl”.

7. Thank you for pointing these out. These are now corrected. Corrected updated lines 197 and 198.

8. In Fig 3b, you mentioned “Representative immunofluorescent micrographs (cold digestion)” line 308, but you have also shown results for hot digestion, please modify.

Thank you for pointing that out. The figure description now indicates both hot and cold digestion. Corrected updated lines 311.

9. What is the magnification used for the images ?

The 20x magnification is now indicated in updated lines 317-318. 

10. What is “Top scale bar, 2mm”, line 310?

Thank you for pointing that out. The “Top scale bar, 2mm” is now removed. The figure description was corrected and indicates a single scale bar of 0.1mm. Updated lines 316. 

11. In Fig 3c, could you specify if the images concern cold or hot digestion condition?

The Fig 3c description now specifies that it concerns cold digestion condition. Updated line 320. 

12. “Militenyi” should be replaced by “Miltenyi”.

Thank you for pointing that out. The typo was corrected throughout the manuscript and S4 supplemental protocol. 

13. In “recipes”, “4C°” should be replaced by “4°C”.

The typo was corrected. Thank you for pointing it out.

14. Please check step 28. Is the cell pellet resuspended in 100µl of airway epithelial growth media (as indicated in supplement S4 protocol) or 0.5ml of MACS buffer (as indicated in Materials and Methods).

Thank you for pointing out the discrepancy. Both the S4 protocol and the manuscript were corrected to indicate 0.5ml of supplemented airway epithelial growth media. Updated line 136.

15. Please add the coating step description before step 30.

Following step is now present in the protocol: “30. Coat the well plate with well-plate coating solution and incubate for at least 4-8h at 37°C before use.”

16. Chemical product names are still not correct. Please check thoroughly all your files (manuscript, supplementary…). CHIR99021 from stemcell Tech should not be CHIR9902 or CHIR9901 (reviewed manuscript). A 83-01 from stemcell Tech should not be A8301 in S3 Fig (manuscript, line 391).

We apologies for these omissions. We have now corrected all indicated product name typos. We also reviewed both the manuscript and the supplementary file with following changes applied:

• Manuscript line 150 changed DMH-1 to DMH1

• Manuscript line 152 changed CHIR9901 to CHIR99021

• Manuscript line 391 changed CHIR9902 to CHIR99021

• Manuscript line 391 changed A8301 to A 83-01

• Manuscript line 151 changed “Y27632” to “Y-27632”

• Manuscript line 391 changed “Y27632” to “Y-27632”

• S4 protocol – DMH-1 changed to DMH1 in material list and in recipes

• S4 protocol – changed CHIR9901 to CHIR99021 in materials list

• S4 protocol – changed CHIR9901 to CHIR99021 in recipes

• S4 protocol – changed “Y27632 ROCK” to “Y-27632” in recipes

17. Do you really have 1% Pen/Strep in your MACS buffer (S4 protocol) ? It is not indicated in “Murine lung harvest and digestion” part (manuscript, line 118).

In addition, in “Recipes” of S4 protocol, you should name this buffer “MACS buffer” instead of “MACS buffer wash”. It is confusing.

We apologies for omitting Pen/Strep in the manuscript MACS buffer recipe. This is now corrected in the manuscript line 118 where we added “1% v/v Penicillin/Streptomycin (10,000 U/ml, Gibco)”. 

We apologise for confusing the reader by using “MACS buffer” and “MACS buffer wash” interchangeably. Naming was corrected and “MACS buffer WASH” in S4 supplement was changed to “MACS buffer”. 

18. In “Recipes” of S4 protocol, you did correct as requested the “4C⁰” in “Airway epithelial growth media“ part, but you have forgotten to correct the “4C⁰” in “Well-plate coating solution “ part. Please modify.

We apologise for the incorrect “4C⁰” in “Well-plate coating solution“ S4 supplement. This has been now corrected to “4C⁰”.

19. Thanks for adding as requested the coating step (step 30) in S4 protocol. However, it would have been better if you had not made a typo. “coate” should be replaced by “coat”.

We apologise for the typo. “coate” is now corrected to “coat”. 

We warmly thank the Reviewer for their positive appraisal, which together with their constructive comments and suggestions have greatly helped us in submitting an improved manuscript to your Journal. We hope this will be received positively and look forward to your decision in due course.

Yours Faithfully,

Professor Jürgen Schwarze

Corresponding Author

---

## [Editor Report · Decision Letter 2]

9 Jan 2024

Cold dispase digestion of murine lungs improves recovery and culture of airway epithelial cells

PONE-D-23-26951R2

Dear Dr. Schwarze,

We’re pleased to inform you that your manuscript has been judged scientifically suitable for publication and will be formally accepted for publication once it meets all outstanding technical requirements.

Kind regards,

Dominique Heymann, Ph.D.

Academic Editor

PLOS ONE
---

## [Editor Report · Acceptance letter]

17 Jan 2024

PONE-D-23-26951R2 

PLOS ONE

Dear Dr. Schwarze, 

I'm pleased to inform you that your manuscript has been deemed suitable for publication in PLOS ONE. Congratulations! Your manuscript is now being handed over to our production team.

Kind regards, 

on behalf of

Pr. Dominique Heymann 

Academic Editor

PLOS ONE